# Antiviral Effect of Candies Containing Persimmon-Derived Tannin against SARS-CoV-2 Delta Strain

**DOI:** 10.3390/v15081636

**Published:** 2023-07-27

**Authors:** Ryutaro Furukawa, Masahiro Kitabatake, Noriko Ouji-Sageshima, Dai Tomita, Makiko Kumamoto, Yuki Suzuki, Akiyo Nakano, Ryuichi Nakano, Yoko Matsumura, Shin-ichi Kayano, Hisakazu Yano, Shinji Tamaki, Toshihiro Ito

**Affiliations:** 1Department of Immunology, Nara Medical University, Kashihara 6348521, Japan; rfurukawa-tky@naramed-u.ac.jp (R.F.); mkita@naramed-u.ac.jp (M.K.); osage@naramed-u.ac.jp (N.O.-S.); y.matsumura@kio.ac.jp (Y.M.); 2Department of Respiratory & Internal Medicine, National Hospital Organization Nara Medical Center, Nara 6308053, Japan; tomita.dai.mg@mail.hosp.go.jp (D.T.); mkuma@naramed-u.ac.jp (M.K.); tamaki.shinji.yw@mail.hosp.go.jp (S.T.); 3Department of Microbiology and Infectious Diseases, Nara Medical University, Kashihara 6348521, Japan; suzuki-y@naramed-u.ac.jp (Y.S.); akiyo@naramed-u.ac.jp (A.N.); rnakano@naramed-u.ac.jp (R.N.); yanohisa@naramed-u.ac.jp (H.Y.); 4Department of Health and Nutrition, Faculty of Health Science, Kio University, Koryo 6350832, Japan; s.kayano@kio.ac.jp; 5MBT (Medicine-Based Town) Institute, Nara Medical University, Kashihara 6348521, Japan

**Keywords:** SARS-CoV-2, COVID-19, Delta variant, persimmon-derived tannin, saliva, aerosol infection

## Abstract

Inactivation of severe acute respiratory syndrome coronavirus 2 (SARS-CoV-2) in the mouth has the potential to reduce the spread of coronavirus disease 2019 (COVID-19), due to the virus being readily transmitted by dispersed saliva. Persimmon-derived tannin has strong antioxidant and antimicrobial activity owing to its strong adhesion to proteins, and it also exhibited antiviral effects against non-variant and Alpha-variant SARS-CoV-2 in our previous study. In this study, we first demonstrated the antiviral effects of persimmon-derived tannin against the Delta variant of SARS-CoV-2 in vitro via the plaque assay method. We then examined the effects of candy containing persimmon-derived tannin. Remarkably, the saliva samples provided by healthy volunteers while they were eating tannin-containing candy showed that the virus titers of the SARS-CoV-2 Delta variant were suppressed. In addition, we found that the SARS-CoV-2 viral load in saliva from patients with COVID-19 collected immediately after they had eaten the tannin-containing candy was below the level of detection via PCR for SARS-CoV-2. These data suggest that adding persimmon-derived tannin to candy and holding such candy in the mouth is an effective method for inactivating SARS-CoV-2 in saliva, and the application of this approach shows potential for inhibiting the transmission of COVID-19.

## 1. Introduction

Coronavirus disease 2019 (COVID-19), which is caused by severe acute respiratory syndrome coronavirus 2 (SARS-CoV-2), remains a serious global threat [1,2,3]. The transmission of this virus from patients with no or mild symptoms by means of dispersed saliva poses one of the greatest challenges to ending the ongoing multi-year pandemic [4,5].

Tannins are a kind of plant-derived polyphenol which are found in a variety of foods and have long been used in different industries. Due to their phenolic hydroxyl groups, tannins have a strong astringency, or protein binding ability, which leads to their antioxidant and antiseptic effects [6]. A persimmon (*Diospyros kaki*) is an edible fruit that contains a large amount of persimmon tannins, which are soluble polyphenols with four types of catechin (epicatechin, epicatechin gallate, epigallocatechin, and epigallocatechin gallate) condensed into chains of 19–47 molecules that are long and highly polymerized [6]. Because of its numerous phenolic hydroxyl groups, persimmon-derived tannin has strong antioxidant, immunomodulatory, antimicrobial, and antiviral activities [6,7,8,9]. A previous report showed that persimmon tannin has stronger antiviral activities against a variety of viruses than other plant-derived tannins, such as green tea tannin or acacia tannin [9]. We previously showed that persimmon-derived tannin inactivated SARS-CoV-2 in vitro and that administering persimmon tannin into the oral cavity of a hamster model reduced the severity and transmission of SARS-CoV-2 infection [10]. Persimmon-derived tannin may be a valuable candidate for use in COVID-19 protection and medical therapy.

In August 2021, the Delta variant of SARS-CoV-2 (B.1.617.2) became the predominant strain of SARS-CoV-2 in Japan. The Delta variant was first identified in England in March 2021, and it rapidly spread around the world. The Delta variant is highly contagious, and several large studies have shown that the Delta variant causes more severe disease compared with the other identified SARS-CoV-2 variants, including the Alpha and Omicron variants [11,12,13,14]. A study in Canada reported that the Delta variant caused more than twice the number of hospital admissions, ICU admissions, and deaths compared with the Alpha variant [11]. Additionally, another study in England [13] revealed that the risk of hospital admission and death in patients infected with the Delta variant was more than double that of patients infected with the Omicron variant. Salivary glands are important sites for the replication of SARS-CoV-2; therefore, the inactivation of virus in the saliva may inhibit droplet and aerosol transmission of SARS-CoV-2 [5].

In this study, we evaluated the antiviral effects of persimmon-derived tannin against the Delta variant of SARS-CoV-2 in human saliva. Herein, we used tannin-containing hard candy to deliver and retain persimmon-derived tannin in the oral cavity. Healthy volunteers and patients with COVID-19 were instructed to eat this candy and then salivate.

## 2. Materials and Methods

### 2.1. Tannin Preparation

The persimmon-derived tannin used in this study was extracted from persimmon fruits harvested in Nara Prefecture in 2011 in accordance with a previously described method [10]. Immature persimmon fruits (*Ebenaceae Diospyros kaki* Thunb., cv. ‘Hiratane-nashi’ and ‘Tonewase’) were treated with 0.2% ethanol for 5 days to insolubilize the tannin. Ethanol-treated persimmons were crushed into small particles and immersed in water for 2 days at room temperature. The resulting supernatant liquid contained soluble components, such as sugars, while the remaining residue contained insoluble tannins. After the supernatant liquid was discarded, water was added to the residue. The insoluble tannins were heated at 120 °C for 30 min to change them into soluble tannins, which were then extracted with water. The extracted soluble tannins were filtered, evaporated in vacuo, and dried at 160 °C. The resulting batch of soluble tannin powder contained 75.5% condensed tannin in terms of epigallocatechin gallate, as assessed using the Folin–Ciocalteu method. The persimmon tannin powder was provided by Ishii-Bussan Inc. (Nara, Japan) and stored at −20 °C until use. The cultivation and harvesting of persimmons by the Ishii Bussan Corporation were authorized by the Agricultural Committee of Tenri City, Nara Prefecture, Japan. All procedures were conducted in accordance with the World Health Organization (WHO) Guidelines on Good Agricultural Practices and Collection Practices (GACP) for Medicinal Plants.

### 2.2. Candies

The candy used in this study was manufactured by Kanro Inc. (Tokyo, Japan) and weighed approximately 4 g per piece. The tannin-containing candy comprised 97% sugar and maltose syrup, 2% persimmon-derived tannin, and 1% other (flavorings, fats, and emulsifiers), and the control candy was composed of 98% sugar and maltose syrup, 1% coloring, and 1% other (flavorings, fats, and emulsifiers). Both the tannin-containing and tannin-free candies were colored dark brown. 

### 2.3. Virus Preparation and Biosafety

The Delta variant of SARS-CoV-2 (hCoV-19/Japan/TY11-927/2021) was isolated and provided by the National Institute of Infectious Diseases, Japan. A virus culture was performed using VeroE6/TMPRSS2 cells (JCRB Cell Bank in Japan, JCRB1819). The virus culture fluid was subjected to two cycles of freezing and thawing, and then it was centrifuged at 10,000× *g* for 15 min at 4 °C. The resulting supernatant was ultra-filtered using an Amicon Ultra-15 (Merck, Darmstadt, Germany) and washed with phosphate-buffered saline (PBS) to prepare the virus extracts. The resulting virus stocks were titrated to determine the number of plaque-forming units (pfu) in VeroE6/TMPRSS2 cells by using the plaque assay method described in Section 2.6. The viral load was measured via quantitative polymerase chain reaction (qPCR) for viral antigens, as described in Section 2.7. All experiments using SARS-CoV-2 were performed at the biosafety level (BSL) 3 experimental facility at Nara Medical University, Japan.

### 2.4. Healthy Volunteers

Three healthy volunteers were invited to participate in the study. All study participants provided written informed consent. The eligibility criteria were that participants had to be ≥20 years of age and have no medical history. The Nara Medical University Ethics Committee approved the study (No. 2846). After they gargled and brushed their teeth, healthy volunteers were instructed to salivate before eating the provided experimental or control hard candy. They then sucked on a tannin-free candy for 10 min before sucking on a tannin-containing candy for 10 min. Saliva samples were collected before and while the volunteers were eating each candy. The saliva samples were mixed with the virus solution containing 1.0 × 10^7^ pfu/mL at a ratio of 9:1. After 10 min of reaction time, bovine serum albumin (Fujifilm Wako, Osaka, Japan), adjusted to 1% using PBS, was added to block the tannin effect. The virus titers of each solution were measured using the plaque assay method described in Section 2.6. PBS was used as the negative control.

### 2.5. Patients with COVID-19

Three patients with mild cases of COVID-19 who were admitted to the National Hospital Organization Nara Medical Center in Nara, Japan, between 28 July and 23 August 2021 were invited to participate in this study. All study participants provided written informed consent. The eligibility criteria were that participants had to be ≥20 years of age and have no major underlying disorders. The study was approved by the Ethics Committees of both Nara Medical University (No. 3012) and the National Hospital Organization Nara Medical Center (No. 2021-3). We collected saliva from three patients with COVID-19 who were diagnosed on the basis of positive SARS-CoV-2 PCR results. They had only mild symptoms. The classification of the severity of symptoms of COVID-19 was based on the treatment guidelines of COVID-19 [15]. Here, patients with mild symptoms had any of the various signs and symptoms of COVID-19, but did not have shortness of breath, dyspnea, or abnormal chest imaging.

Eight saliva samples were collected from each patient. Patients were instructed to produce 2 mL of saliva before, immediately after, 15 min after, and 60 min after they finished eating the tannin-free control candy for 10 min. They then salivated in the same way while eating the tannin-containing candy. The bovine serum albumin adjusted to 1% using PBS was added to the samples to block the tannin effect immediately after sampling. RNA was extracted from each saliva sample, and the SARS-CoV-2 viral load was determined by detecting the nucleocapsid gene of the virus via qPCR, as described in Section 2.7.

### 2.6. Determination of Virus Titers Using the Plaque Assay

The titers of infectious SARS-CoV-2 were measured using the plaque assay, as previously described [10]. Briefly, VeroE6/TMPRSS2 cells were grown to confluence on 12-well plates. Samples were serially diluted tenfold with PBS and administered to the cells. After a 1 h inoculation, the cell monolayer was covered with an agarose overlay (Nippon Gene, Tokyo, Japan) and incubated for 48 h at 37 °C with 5% CO_2_. After the agarose overlay was removed, the cells were fixed with 10% formalin and then stained with crystal violet. The number of visible plaques was counted, and the number of pfu in each solution was determined. The active virus titers are expressed as the pfu/mL of the solution.

### 2.7. qPCR

The QIAamp Viral RNA Mini Kit (Qiagen, Venlo, The Netherlands) was used to extract RNA from patient saliva, and the RNA was reverse-transcribed to cDNA using a High-Capacity cDNA Reverse Transcription Kit (Thermo Fisher Scientific, Waltham, MA, USA) in accordance with the manufacturer’s instructions. The SARS-CoV-2 viral load was determined by detecting the nucleocapsid gene of the virus via qPCR. As recommended by the manual prepared by the National Institute of Infectious Diseases, Japan, we performed qPCR for SARS-CoV-2 using the TaqMan FAST Advanced Master Mix with the following primer set: forward primer, 5′-AAATTTTGGGGACCAGGAAC-3′; reverse primer. 5′-TGGCAGCTGTGTAGGTCAAC-3′; and the TaqMan probe, FAM-ATGTCGCGCATTGGCATGGA-BHQ. The numbers of RNA copies in saliva samples were calculated from the CT values measured via qPCR by comparing them to those of a standardized control.

The N501Y, E484K, L452R, and E484Q mutations of the SARS-CoV-2 spike protein were determined via qPCR using mutation detection kits purchased from Takara Bio (Kusatsu, Japan; product numbers RC344A, RC345A, RC346A, and RC349A, respectively). SARS-CoV-2 that was positive for L452R and negative for N501Y was considered to be the Delta variant; however, we did not perform a more detailed examination, such as genome sequencing.

### 2.8. Statistical Analysis

Data are presented as the mean ± SEM and are representative of at least two independent experiments. Statistical analyses were performed via Student’s *t*-tests using GraphPad Prism 6 (GraphPad Software, San Diego, CA, USA). *p*-values of <0.05 were considered to indicate a statistically significant difference.

## 3. Results

### 3.1. Persimmon-Derived Tannin Inactivates the Delta Variant of SARS-CoV-2 In Vitro

First, to investigate the antiviral activity of persimmon-derived tannin against the Delta variant of SARS-CoV-2 in vitro, we examined the replicative ability of the Delta variant reacting with persimmon-derived tannin. The results show that the persimmon-derived tannin inactivated the virus in a concentration- and exposure time-dependent manner (Figure 1). These data are consistent with our previous report, in which we demonstrated the antiviral activity of persimmon-derived tannin against non-variant and Alpha-variant SARS-CoV-2 [10].

### 3.2. Saliva Sampled While Eating Persimmon-Derived Tannin-Containing Candy Inactivates the Delta Variant of SARS-CoV-2 In Vitro

Next, we investigated the antiviral effects of saliva collected from three healthy volunteers before or after sucking on a persimmon-derived tannin-containing hard candy for 10 min. Hard candy is a good system for spreading tannins in the oral cavity. The antiviral effects of saliva from each volunteer are shown in Figure 2. Saliva collected either before or after eating the control candy did not suppress the virus, whereas saliva collected after eating the persimmon-derived tannin-containing candy significantly suppressed the virus titers by approximately 1/1000.

### 3.3. Administering Persimmon-Derived Tannin-Containing Candy to Patients with COVID-19 Caused by SARS-CoV-2 Delta Variant Can Suppress the Viral Load in Their Saliva

We then conducted a clinical examination using the same tannin-containing hard candy. Three patients with COVID-19 and mild symptoms, who had no major underlying disorders, were administered the candy, and their saliva was sampled (Figure 3). RNA was extracted from each sample, and qPCR was performed to detect the nucleocapsid gene of SARS-CoV-2. Viruses from all three patients were found to be positive for the L452R mutation and negative for the N501Y mutation, indicating that the patients were infected with the Delta variant of SARS-CoV-2 [16]. The results show that the saliva collected immediately after eating the tannin candy was negative for SARS-CoV-2, as assessed via PCR (Figure 4). The viral load in saliva then gradually increased over time, but it was still suppressed to approximately 1/1000 of that of the control even 60 min after finishing the tannin candy. In contrast, saliva collected while eating the control candy did not exhibit such antiviral effects (Figure 4).

## 4. Discussion

First, we clearly demonstrated that persimmon-derived tannin inactivated the Delta variant of SARS-CoV-2 in vitro. Such an antiviral effect against the Delta variant, which has caused the most severe cases of COVID-19 compared with other known SARS-CoV-2 variants, is highly beneficial. The observed effect was dose- and time-dependent, and it was quite similar to the findings from our previous report using non-variant and Alpha-variant SARS-CoV-2 [10]. Persimmon-derived tannin may inactivate SARS-CoV-2, regardless of the variant type. This feature of persimmon-derived tannin is completely different from what has been observed with vaccines and neutralizing antibody therapy, which have become less effective against SARS-CoV-2 variants such as Delta and Omicron [17,18], suggesting that persimmon-derived tannin could be effective even against new SARS-CoV-2 variants that may appear in the future. 

The mechanism of the antiviral effect of persimmon tannin is partly understood. In our previous report, we demonstrated that persimmon tannin adheres directly to SARS-CoV-2 and largely aggregates with the virus due to its strong protein binding ability [10]. The virus may be totally covered with persimmon tannin, and therefore loses its infection ability. This mechanism might explain the reason why the antiviral effect of persimmon tannin is not affected by the virus variant. Pentarlandir ultrapure and potent tannic acid (UPPTA) is reported to prevent the replication of SARS-CoV-2 by acting on the 3-chymotrypsin-like protease (3CLpro) in VeroE6 cells and being distributed in the lungs, which is the primary site of SARS-CoV-2 infection [19]. A clinical study using a tannin-based supplement did not result in the clinical improvement of COVID-19 patients but showed a reduction in the inflammatory state and microbiota modulation [20]. Our previous study also showed oral persimmon tannin supplementation for the murine model of *Mycobacterium avium* complex infection attenuated the inflammatory status and pathogenesis of the pulmonary lesion [8]. Although not considered in our current study, candies containing persimmon-derived tannin might have similar antiviral and anti-inflammatory effects. The mechanisms of the distribution and absorption of tannin are not understood either. Highly polymerized persimmon tannin is not considered to absorb well and is broadly distributed when administered in the oral cavity, and therefore our current study did not include the distribution of tannin in the body, such as, for example, in the lungs; however, it might be important to examine the effects of tannin in places other than in the oral cavity. In addition, there are no reports of drug–drug interactions between the persimmon-derived tannin and commonly prescribed medicines as far as we found, but this should be considered. Further investigations are needed to clarify these issues.

We also demonstrated that human saliva obtained from a volunteer while they were eating tannin-containing hard candy could inactivate SARS-CoV-2, and that eating tannin-containing hard candy reduced the viral load in the saliva of patients with COVID-19. Saliva from volunteers significantly reduced the virus titers of SARS-CoV-2 by approximately 1/1000. Persimmon-derived tannin shows antiviral effects even when it is mixed with human saliva. In addition, our clinical examination revealed that saliva from patients with COVID-19 caused by the Delta variant that was collected just after eating tannin-containing candy produced negative results when tested by SARS-CoV-2 PCR, and the virus inhibition effect lasted for at least 60 min. Tannins administered into the oral cavity are expected to remain there because they easily adhere to cells and viruses. Our current results indicate that the oral administration of persimmon tannin may remain effective for a reasonably long time in clinical situations. 

Reducing the SARS-CoV-2 viral load in saliva may suppress the risk of COVID-19 transmission in humans. SARS-CoV-2 mainly causes lung diseases, but the oral cavity is also an important location of virus infection. The salivary gland is known as one of the main areas for virus replication, and a high viral load of SARS-CoV-2 is often detected in saliva [5]. Interestingly, not only do symptomatic patients have a high viral load of SARS-CoV-2 in their saliva and are highly contagious, but asymptomatic patients are as well [21]. Saliva droplets are generated when breathing and talking, as well as coughing and sneezing, and can spread several meters from the infected person [22]. That means saliva is easily dispersed in situations such as face-to-face conversation and dining. Actually, restaurant dining has been closely associated with COVID-19 case and death growth rates [23]. Although respiratory droplets/aerosol transmission from coughs and sneezes can be the most significant route for community transmission, dispersed saliva from both symptomatic and asymptomatic patients may be greatly involved in the transmission of COVID-19 in daily life. In addition, in our previous study using an in vivo hamster model [10], the transmission of SARS-CoV-2 infection from infected hamsters to non-infected hamsters was suppressed by administering persimmon-derived tannin into the mouth of infected hamsters. Our current study supports the potential ability of persimmon-derived tannin administered in the mouth to inhibit COVID-19 transmission in humans. 

We used a hard candy to administer the persimmon-derived tannin into the mouth, even though mouthwashes have been more often used for similar purposes in previous studies [24]. The concentration of persimmon-derived tannin used in this study, 2%, was determined from our previous report [10], which showed that 20 mg of tannin powder mixed with 100 μL of solvent (=2%) that was administered into the oral cavity in an in vivo hamster model effectively prevented severe COVID-19 pneumonia due to both direct infection and virus transmission. Hard candy is easily kept in the mouth for several minutes, which can allow the tannin to effectively inactivate the virus in the mouth. Persimmon-derived tannin-containing candy can be used as a simple, safe, inexpensive, and immediately available method of COVID-19 prophylaxis, and it has the potential to be helpful in fighting against the ongoing COVID-19 pandemic.

However, our study has several limitations. First, although the antiviral effect of tannin is dose-dependent, we did not assess the tannin concentration attained in the saliva while a tannin-containing candy is being consumed. It is difficult to fully assess the tannin in saliva because the tannin binds to proteins in the oral cavity, and also because the amount of saliva varies among individuals. Second, only three patients with COVID-19 caused by the Delta variant were included in our clinical examination because only hospitalized patients who had a mild form of the disease without complications were enrolled in the study, and because the Delta variant was no longer as prevalent in Japan when the study participants were enrolled. The available data from patients are limited. Third, this study was an open-label test, and all participants ate the control candy first and then the tannin candy. The order of the candy intake could have affected the results. Fourth, we performed plaque assays to assess the patient saliva samples, but visible plaques were not observed. The plaque assay we used may not be sensitive enough to detect virus infectivity when viral loads are low. A previous report similarly found that plaques could not be successfully observed from saliva samples, possibly because plaque formation was inhibited by saliva components [25].

In summary, we clearly demonstrated that persimmon-derived tannin has the potential to inactivate the SARS-CoV-2 Delta variant both in vitro and in vivo. Persimmon-derived tannin is a highly safe product extracted from a natural plant and may be a valuable candidate for COVID-19 protection regardless of the SARS-CoV-2 variant type.

## Figures and Tables

**Figure 1 viruses-15-01636-f001:**
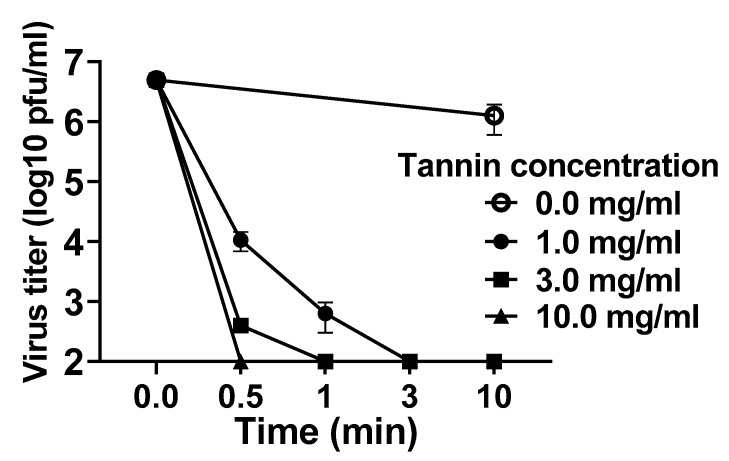
In vitro examination of persimmon-derived tannin against the Delta variant of SARS-CoV-2. Tannin powder was diluted with distilled water and mixed with an amount of the virus solution equal to 1.0 × 10^7^ plaque-forming units (pfu)/mL. After 0.5 min, 1 min, 3 min, and 10 min of reaction time, bovine serum albumin adjusted to 1% using phosphate-buffered saline was added to block the tannin effect. The virus titers of the solution were measured using the plaque assay method. Presented data are the mean ± SEM of two independent experiments.

**Figure 2 viruses-15-01636-f002:**
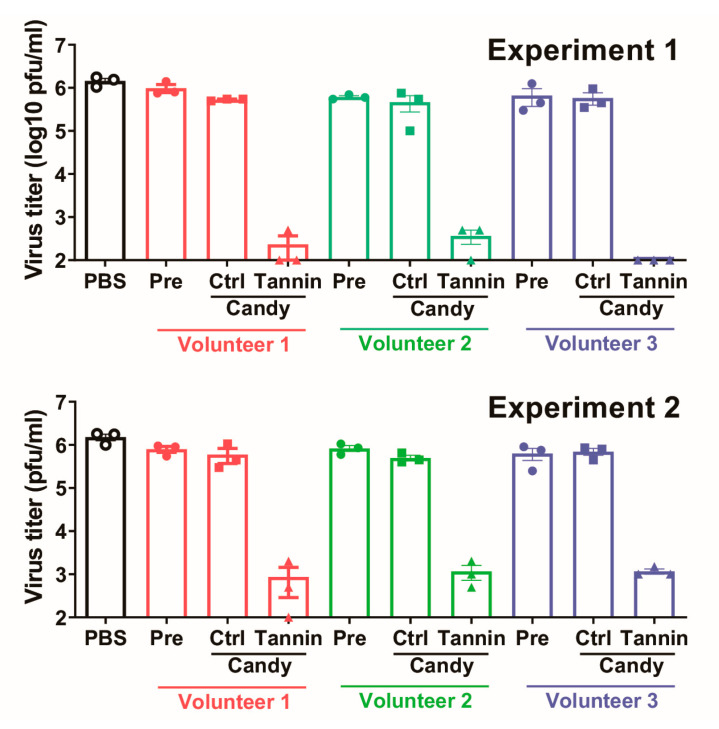
Antiviral effect against the Delta variant of SARS-CoV-2 of saliva collected while eating tannin-containing candy. Saliva samples were collected from three healthy volunteers (Volunteers 1, 2, and 3) before eating candy (Pre), after eating tannin-free control candy (Ctrl), and after eating tannin-containing candy (Tannin). Virus solution containing 1.0 × 10^7^ plaque-forming units (pfu)/mL was mixed with saliva sample or phosphate-buffered saline (PBS) control at a ratio of 1:9. After 10 min of reaction time, bovine serum albumin adjusted to 1% using PBS was added to block the tannin effect. The virus titers of each solution were measured using the plaque assay method. The results of two independent tests (Experiments 1 and 2) are shown. Presented data are the mean ± SEM.

**Figure 3 viruses-15-01636-f003:**
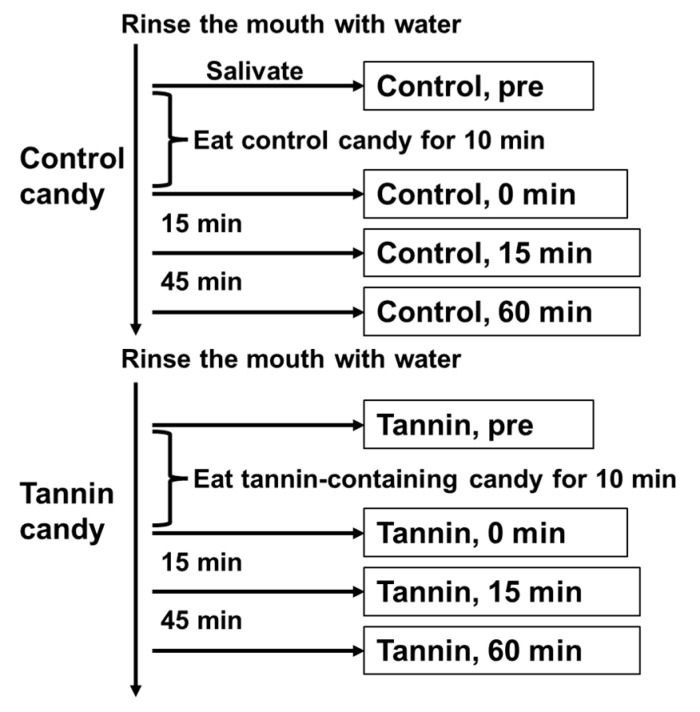
The study protocol for sample collection from patients with COVID-19. Participants were instructed to eat candy (tannin-containing or control) and produce approximately 2 mL of saliva. Eight samples were collected from each participant.

**Figure 4 viruses-15-01636-f004:**
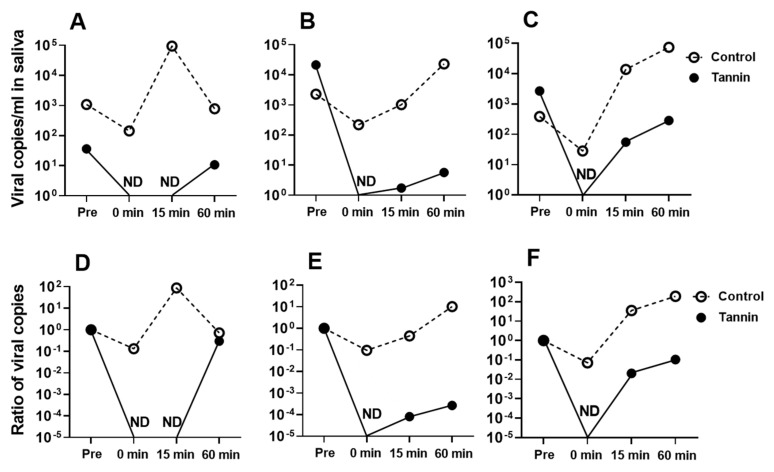
Viral loads in saliva samples from patients with COVID-19 before and after consuming persimmon-derived tannin-containing candy. Saliva samples were collected from three patients with laboratory-confirmed COVID-19 in accordance with the protocol depicted in Figure 3. The viral load in each saliva sample was measured using qPCR. The upper graphs (**A**–**C**) show the numbers of viral copies of SARS-CoV-2 in saliva, and the lower graphs (**D**–**F**) show the ratios of viral copy numbers; the number of viral copies in saliva collected before eating candy (Pre) was used as the standard. The graphs (**A**,**D**) show the data from the first patient, (**B**,**E**) the second patient, and (**C**,**F**) the third patient. ND—not detected.

## Data Availability

Data are contained within the article.

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
