# Peer review of "Antiviral Effect of Candies Containing Persimmon-Derived Tannin against SARS-CoV-2 Delta Strain"

_viruses, 2023, doi:10.3390/v15081636_

Round 1

Reviewer 1 Report

I believe the intend of the manuscript is valid but the manuscript needs some work before it is ready for the show.

1. In lines 34-35, “Coronavirus disease 2019 (COVID-19) caused by severe acute respiratory syndrome coronavirus 2 (SARS-CoV-2) remains a serious, global threat. (DOI: 10.1038/s41579-022-00839-1; DOI: 10.3390/biomedicines9060689; DOI: 10.1038/s41579-023-00878-2)”.

2. In lines 39-45, It is suggested to add more detail about tannin. “Pentarlandir™ UPPTA, a naturally occurring polyphenol with antioxidant properties, potently inhibited SARS-CoV-2 infection by targeting 3CLpro in Vero E6 cells. The EC50, EC90, and EC99 were 0.585, 8.307, and 12.90 μM, respectively. Pentarlandir™ UPPTA showed excellent target distribution in the lungs, which is the primary site of SARS-CoV-2 infection. (Eur. J. Med. Chem. 2023, 257, 115503)”

3. Please explain the mechanism (potential drug targets) of its antiviral effects.

4 The authors seem to ignore the importance of pharmacokinetics in drug discovery. Can a drug be distributed to the virial infection site, such as lung for SARS-CoV-2? Will it have drug-drug interactions with other commonly prescribed medicines?

Minor editing of English language required

Reviewer 2 Report

The authors report the antiviral effect of persimmon-derived tannin against SARS-CoV-2 (delta variant). They did a very small study with human subjects, showing that 1) saliva of volunteers who ate the tannin-containing candy, has antiviral effect; and 2) in SARS-CoV-2 infected persons, eating the candy leads to lower viral loads in their saliva. They used only three subjects in each study, but the reducing effect on the virus is quite impressive and consistent.

This is an original study that is very limited in scope and size, but still quite intriguing. I have only a few minor comments:

1)   Is the chemical structure of persimmon-derived tannin known? Does it have unique properties compared to other plant tannins?

2)   Lines 235-237. This description/hypothesis of how this tannin works is too vague. Add one or two sentences that summarize the findings in refs 6 and 7.

3)   Fig. 1: the active concentrations of tannin are very high. How does this relate to the concentration in the candy?

4)   Specify a bit the ‘mild symptoms’ of the COVID-19 patient volunteers.

5)   The authors overestimate the importance of saliva in transmission of SARS-CoV-2. Droplet/aerosol transmission is considered to be THE most important route for community transmission, e.g. in poorly ventilated rooms. Please add more nuance when describing the role of saliva, in the introduction and discussion.
